# The Effect of Temperature and Humidity on the Filtration Performance of Electret Melt-Blown Nonwovens

**DOI:** 10.3390/ma13214774

**Published:** 2020-10-26

**Authors:** Chao Liu, Zijian Dai, Bin He, Qin-Fei Ke

**Affiliations:** 1Key Laboratory of Textile Science & Technology of Ministry of Education, College of Textiles, Donghua University, Shanghai 201620, China; zjdai@dhu.edu.cn; 2Textile and Fashion Collage, Hunan Institute of Engineering, Xiangtan 411101, China; 70210@hnie.edu.cn

**Keywords:** filtration efficiency, melt-blown nonwovens, charge decay, heat, heat-moisture

## Abstract

Electret melt-blown nonwovens are widely used for air purification due to their low pressure drop and high filtration efficiency. However, the charge stability could be affected by the ambient temperature and humidity, reducing the filtration efficiency, resulting in the electret melt blown filter not providing effective protection. Herein, we used corona charge to prepare electret melt-blown nonwovens and systematically studied the effects of different temperature and humidity on the structure, morphology, filtration performance, and surface potential within 24 h. The effect of treatment temperature and humidity on pressure drop was minimal because the fiber morphology and web structure of melt-blown nonwovens were not damaged. When the treatment temperature was lower than 70 °C, the effect on the filtration efficiency of the sample was small, but when the temperature increased to 90 or 110 °C, the filtration efficiency decreased significantly with the increase of the treatment time, and the surface potential also declined similarly. In conclusion, high temperatures will lead to charge escape and reduce the electrostatic adsorption effect. Furthermore, at the same temperature, increasing relative humidity can accelerate the charge release and make the filtration efficiency drop more. After the sample was treated at 110 °C and 90% relative humidity for 24 h, the filtration efficiency decreased from 95.49% to 38.16% at a flow rate of 14.16 cm s^−1^, and the surface potential dropped to the lowest value of −1.01 kV. This result shows that all links of electret melt-blown filter material from raw material to final use should be avoided in high temperature and high humidity conditions to ensure the protection effect.

## 1. Introduction

Airborne fine particulate pollution has serious direct effects on public health. Many epidemiological studies have reported that exposure to particulate matter with an aerodynamic diameter less than 2.5 μm (PM2.5) could increase cardiopulmonary risk [1,2,3,4,5]. Presently, the simplest and most common protection method is the use of air filters. Melt-blown nonwovens have become the preferred air filter materials due to their three-dimensional random arrangement of fibers, large specific surface area, high porosity, small pore size, and high barrier performance [6,7]. It is widely used to prepare protective masks, fresh air systems, air purifiers and other air filters. The filtration mechanism of fiber materials for airborne particles is very complicated. There are many related theoretical studies, among which the classical filtration theory mainly focuses on the single fiber filtration mechanism. It is currently believed that in the steady-state filtration stage, the capture of particles on the single fiber follows one or a combination of the following mechanisms: inertial impaction, direct interception, gravity settling, diffusion, and electrostatic attraction (see Figure 1) [8]. The particle size ranges corresponding to different filtration mechanisms can be approximately allocated, as shown in Figure 2. Particles larger than 10 μm deviate from the air streamline before reaching the filter because of gravity settling. Relatively larger size particles (greater than 0.3 μm) are filtered by the interception and inertial impaction [9]. Microparticles (less than 0.1 μm) are effectively filtered by diffusion is based on the random (Brownian) motion of particles that will make the particle deviate from the air streamline and collide with fibers. The diffusion effect is more significant when the flow rate is lower and particles are smaller [10].

Electrostatic attraction occurs when the electrostatic or electric charge on the particles and/or fibers will force the particle to divert from the air streamline and attract the fiber [11,12,13]. Electrostatic attraction plays an important role in electret melt-blown air filter material. Due to the ability to store a large number of electric charges and generate a quasi-permanent electric field around the fibers, electret melt-blown nonwovens have been proved to be an ideal material to filter PM2.5 pollution by the electrostatic effect [14,15,16]. The corona charged electret has been often used in polypropylene (PP) melt-blown materials due to its advantages of operability at normal temperature and pressure, high charge density and applicability to all electret fiber materials [17,18]. The filtration performance of electret filters is largely determined by the electrostatic removal mechanism [19]. PP melt-blown electret filters possess excellent space charge storage stability at room temperature. However, the heat setting process of melt-blown filter products, hot and humid storage, and transportation conditions may cause charge attenuation, reducing the electret effect and affecting the final products’ filtration performance.

In this study, we first prepared the electret melt-blown nonwovens by corona charge. Then treated the samples in heat or heat-moisture conditions to systematically study the effect of different temperatures, humidity, and treatment time on the structure, morphology, filtration performance, and surface potential of electret melt-blown nonwovens. The results can provide scientific reference and basis for temperature and humidity control during production and transportation of products, minimize electrostatic charge decay, and ensure that the melt-blown filter products provide efficient protection during use.

## 2. Materials and Methods

### 2.1. Materials

Polypropylene resins (PP, Melt flow index 1500) were purchased from Dawn Co. (Yantai, Shandong Province, China).

### 2.2. Preparation of Melt-Blown Nonwoven

Melt blown nonwoven with a basis weight of 25 g/m^2^ was produced by using a 1600 melt blown line (Changshu Weicheng Nonwoven Fabric Equipment Co., Ltd., Changshu, China). As shown in Figure 3, the solid PP resins were first supplied into the screw extruder through a hopper. The resin was gradually melted by the heating screw treatment and extruded out through the polymer orifice. Then, the extruded polymer stream was attenuated into microfibers by the high-velocity hot air. In the meantime, the same air streams deliver the fiber to the collector, and the bonding took place at the fiber-to-fiber contact points. Finally, the fibers were cooled at room temperature to form the nonwoven. The technological parameters applied in the melt-blown process were set as follows. The screw extruder was heated in seven separate regions, correspondingly, the seven parts of the screw extruder temperatures were maintained to 170, 190, 210, 220, 230, 235, and 235 °C. It is equal to that the temperature of the extruded polymer melt was 230 °C. Air with a pressure of 0.4 MPa and a temperature of 240 °C was applied to the melt-blown spinneret. For the spinneret structure, the apex-angle of the air plates was 60°, and the diameter of the polymer orifice was 0.3 mm. All the microfibrous nonwovens were collected on the forming belt, and the minimum die to collector distance was 180 mm. Notably, to improve the filtration performance of the melt-blown nonwovens, corona charging treatment was applied to eject electrostatic charges into the polymeric fibers. In this article, we employed molybdenum wire to conduct negatively corona charging at a voltage of 100 kV on both sides of the melt-blown nonwovens with a charging distance of 110 mm. Finally, the melt-blown nonwovens were wound into a roll by a winder.

### 2.3. Heat and Heat-Moisture Treatment of Melt-Blown Nonwovens

The influence of temperature and relative humidity on the filtration performance of electret melt-blown air filter materials were studied. Firstly, the pre-cut melt-blown samples (20 cm × 20 cm) were put into the constant temperature blast drying oven (101A-23, Shanghai Thorpe Instrument Co., Ltd., Shanghai, China), then heat treated at four temperatures (50, 70, 90, and 110 °C) and various times (1–24 h). Furthermore, to simulate the hot and humid environment, the pre-cut melt-blown samples were put into a constant temperature and humidity box (YM-33, Laizhou Yuanmao Instrument Co., Ltd., Laizhou, China), the RH was set to 90%, and different heat treatment temperatures and times were set.

### 2.4. Characterization

The surface morphology of the melt-blown nonwovens was observed by using JSM-IT300A (JEOL Ltd., Tokyo, Japan) analytical scanning electron microscope (SEM) after a thin AU-Pd layer was sputter-coated on samples. The Image-Pro Plus software measured the fiber diameter. The average diameter was calculated by at least 100 measured diameters. Moreover, the pore size of the melt-blown nonwovens was characterized by a capillary flow porometer (CFP-1100AI, Porous Materials Inc., Ithaca, NY, USA) based on the bubble point test.

The filtration performance of melt-blown nonwovens was tested by an automatic filter tester (Model 8130, TSI Inc., Shoreview, MN, USA), generating sodium chloride (NaCl) aerosol with a mean particle size of 0.26 μm at a geometric standard deviation of 1.86. The particle size distribution of the aerosol ranges from 0.1~2 μm. The aerosol particles passed through the samples with a valid test area of 100 cm^2^ at a preset flow rate. The air pressure drop for the filters was measured by a flow gauge and two electronic pressure transmitters. More details about the test equipment are shown in Figure 4. Different air flow rates were set to test the filtration performance of samples, and each sample was tested three times to reduce random error. The upstream and downstream photometers are used to measure the aerosol quantities on both of the upstream and downstream sides of the filters. Then the filtration efficiency can be calculated as follows in Equation (1) [20,21].
(1)$$FE^{}(\%)=\left(1-\frac{C_{d}}{C_{u}}\right)\times 100$$
where *C_d_* and *C_u_* represented the downstream and upstream quantities of NaCl aerosol, respectively.

Quality factor (*QF*) is usually used to characterize the filtration performance of filter materials [22,23,24], and it could be calculated by Equation (2).
(2)$$QF=-\frac{ln\;(1-\eta )}{\Delta_{}p}$$
where *η* is the filtration efficiency and Δ*p* is the pressure drop.

The surface electrostatic potential of electret melt-blown nonwovens was measured by a vibrating electrode with compensation employing a non-contacting electrostatic probe (TREK-542A-2-CE, TREK Inc., Lockport, NY, USA). Firstly, the sample was placed on an insulating triangular support. Secondly, install the probe on the insulating triangular support, and keep the distance of 1.5 cm to the samples. Finally, turn on the power, start the test. For each sample, three different areas should be tested. The test process was conducted at an ambient temperature of 25 ± 2 °C and relative humidity of 50 ± 5% for all samples.

## 3. Results

### 3.1. Structure and Performance of the Melt-Blown Nonwovens

Figure 5a–c shows the morphologies of the melt-blown nonwovens characterized by SEM. A randomly arranged three-dimensional microporous structure could be observed on melt-blown nonwovens, making it an appropriate candidate for air filtration. Compared with the untreated melt-blown sample, there is no visible change in appearance and structure of the samples after heat treatment at 110 °C for 24 h (110 °C-24 h), and heat-moisture treatment at 110 °C for 24 h under 90% RH (110 °C-24 h-90%), all of the samples show a smooth surface. The basic weight and thickness are shown in Figure 5d, after the heat or heat-moisture treatment, the basic weight of the samples did not change significantly, but the thickness increased slightly. This may be due to the melt-blown nonwovens after winding were pressed by the outer layers, which made the fiber arrangement closer. However, the heat treatment process can eliminate the internal stress of the fiber that is generated during the compression process, and then the compressed fibers were relaxed, the shape recovered, making the web structure more fluffy and increasing the thickness.

Figure 5e displays the average fiber diameter of the pristine PP melt-blown was 2.64 ± 0.69 μm, and the samples after heat or heat-moisture treatment had no obvious change in fiber diameter. This was because the heating temperatures were lower than the melting temperature of polypropylene, and there was no external force during the heating process, so the fiber and web structure of melt-blown nonwovens were not affected. The pore size distribution was further investigated. As shown in Figure 5f, the average pore size ranged from 16.9 to 17.5 μm, indicating that the pore sizes were also not affected by the temperature and humidity. This can be ascribed to the fact that pore sizes are closely related to fiber. The pore size distributions further confirmed the conclusion. All samples possessed uniform and similar pore size distributions.

### 3.2. Relationship between the Filtration Performance and Air Flow Rate

The air flow through the material is an important factor affecting the filtration performance. Based on different application conditions, the filtration performance versus air flow rates (20–100 L min^−1^) was systematically measured using three samples (untreated, 110 °C-24 h, and 110 °C-24 h-90%). As shown in Figure 6a, the filtration efficiency of three samples decrease with the increase of air flow rate, This result could be explained by the reduced retention time of particles in the samples caused by the higher air flow rate, which directly reduces the possibility of particles colliding with the fibers through Brownian diffusion [25,26]. Besides, the filtration efficiency of the untreated sample decreased slowly to 94.77% when the air flow rate is 16.65 cm s^−1^, while the filtration efficiency of 110 °C-24 h and 110 °C-24 h-90% samples decreased significantly to 46.38% and 34.04%, respectively. This is due to the large attenuation of electrostatic charge in the sample after heat treatment, so the sample mainly depends on their fine fiber diameter, small pore size and large specific surface area to filter particles [27], so the increase of flow rate reduces the interception rate of particles. However, the untreated sample still has strong electrostatic adsorption effect, which can capture particles not only by mechanical effect but also by electrostatic attraction. The strong electrostatic force could also capture particles in a short time as the air flow rate increases.

Moreover, the pressure drop of the three samples is an approximately linear increase with increasing the air flow rate (see Figure 6b), which is consistent with Darcy’s law of viscous resistance [28]. The slope of the linear fit between the pressure drop and the air flow rate of the untreated sample is only 4.179, indicating that it has higher air permeability [29]. After heat or heat-moisture treatment, the slope of the linear fit of 110 °C-24 h and 110 °C-24 h-90% samples decreased slightly to 3.918 and 3.734, respectively. The pressure drop of the treated samples was marginally lower than that of the untreated sample when the flow rate was low, but the difference became obvious as the flow rate increased. When the flow rate increased to 14.16 cm s^−1^, the pressure drop of 110 °C-24 h and 110 °C-24 h-90% samples were 56.7 and 55.3 Pa, respectively, which were lower than the 60.7 Pa of the untreated sample. The reason may be that the heat or heat-moisture treatment process made the web structure fluffy and the thickness increased a little, but the basic weight of the web did not change basically (see Figure 5d) so that the porosity increased. The air flow would choose a short and unblocked path through the filter media following the principle of minimum resistance [30], on this basis, increasing the porosity could reduce the resistance. In general, the pressure drop difference between the treated and untreated samples was relatively small, because the entire treatment process did not damage the web structure and fiber morphology, but only a little increase in thickness.

### 3.3. Relationship between the Filtration Performance and the Heat Treatment Conditions

The corona charge process makes the charged ions deposited on the electret material to form surface charges [31] and trapped inside the material to form space charges [32,33]. The space charge in melt-blown PP electret air purification material mainly comes from the capture of interface traps [34]. The space charge can obtain energy from the outside as the external temperature rises, so the charge may escape the trap when the obtained energy is greater than the trap’s binding energy, which will cause the decay of the space charge.

Figure 7a displays the filtration efficiency decay rates of corona charged melt-blown nonwovens as a function of heat treatment duration at four temperatures (viz., 50, 70, 90, and 110 °C), and heat-moisture treatment in relative humidity of 90% at the same temperatures. The melt-blown samples were tested at a common air flow rate of 5.33 cm s^−1^, and the result shows that filtration efficiency decreased as the treatment time increased. However, the depth of filtration efficiency decline varies with processing conditions and time. Figure 5 indicates that the heat and heat-moisture treatment process did not damage the fiber morphology and web structure, so the reason for the decrease in filtration efficiency was that the electrostatic attraction effect decreased due to the charge attenuation. When the heating temperature was 50 and 70 °C, the filtration efficiency of the melt-blown sample hardly decreased in a short time, and the filtration efficiency had a small decline after treatment for 24 h. The filtration efficiency of sample 50 °C-24 h and 70 °C-24 h decreased slightly from 98.46% of the untreated sample to 97.61% and 96.92%, respectively. The reason may be that the temperature was lower, so the charge dispersion in the material was not easy to occur when the treatment time was short. As the treatment time increased, a small amount of surface or shallow trapped charges was released [35], but the effect on filtration efficiency was small due to less charge attenuation. However, the filtration efficiency decreased significantly when the treatment temperature increased to 90 and 110 °C. The filtration efficiency of samples 90 °C-24 h and 110 °C-24 h were 81.02% and 64.02%, respectively. This result may contribute to the high temperature that causes space charges to escape from shallow and deep traps [36]. The amount of charge attenuation increased with the increasing of heat treatment time, so that the filtration efficiency decreased more obviously. Moreover, the filtration efficiency decreased sharply to 86.61% when the sample was treated at 110 °C for 1 h (110 °C-1 h), which indicated that the charge dissipation rate would be faster with the increase of treatment temperature. In addition, when the melt-blown sample was subjected to heat-moisture treatment under 90% RH, its filtration efficiency is relatively lower than that of the sample treated at the same temperature and time. The filtration efficiency of sample 110-24 h-90% exhibited a maximum decrement of 40.45% in the whole testing process. Because some water molecules may enter the interior of the material under high temperature and humidity conditions and become the carrier of the trapped charges, which increases the charge migration and accelerates the decay of charges [37].

Figure 7b shows the pressure drop of melt-blown nonwovens decreased slightly with the increase of treatment time under different conditions. As discussed in Section 3.1, the heat and heat-moisture treatment could make the melt-blown web structure more fluffy, so its resistance was reduced. Furthermore, the pressure drop decreased more at the condition of 90% RH. When the test flow rate was 5.33 cm s^−1^, the pressure drop of the untreated sample is 22.45 Pa, while that of sample 110-24-90% decreased to 19.45 Pa. In general, there is a slight decrease in pressure drop because the fiber morphology and web structure were not damaged during the whole treatment.

As shown in Figure 7c, the QF value displays a roughly similar trend to the filtration efficiency. Because the filtration efficiency of the sample after high temperature treatment (90 and 110 °C) decreased obviously, while the pressure drop changed little. The QF value of untreated sample was 0.186 Pa^−1^, while that of sample 110-24 h-90% was only 0.045 Pa^−1^, which indicates that high temperature and humidity treatment can significantly reduce the filtration performance of electret melt-blown nonwovens.

In addition, 14.16 cm s^−1^ is also a commonly used air flow rate for the filtration performance test of melt-blown air filter materials, so the filtration performance of all samples was also tested at the air flow rate of 14.16 cm s^−1^, as illustrated in Figure 7d–f. Compared with the result measured at a flow rate of 5.33 cm s^−1^, the decreasing trend of filtration efficiency, pressure drop, and QF value with treatment time was basically the same. However, under the same conditions, the filtration efficiency measured at a flow rate of 14.16 cm s^−1^ was lower, and the pressure drop was larger, resulting in lower QF value. The filtration efficiency of the untreated sample is 95.49%, the pressure drop is 60.7 Pa, and the calculated QF value is 0.051 Pa^−1^, whereas the filtration efficiency of sample 110 °C-24 h-90% drops to 38.16%, the pressure drop is 54.3 Pa, and the QF value is only 0.009 Pa^−1^. This result can be attributed to the reduced retention time of particles in the melt-blown materials caused by the higher air flow rate, which directly reduces the possibility of particles colliding with the fibers. Therefore, the appropriate test flow rate should be selected according to the material application and performance requirements to evaluate its filtration performance.

### 3.4. Attenuation of Surface Potential

To further provide insight into the mechanism of filtration efficiency decay, we systematically evaluated the charge attenuation of melt-blown samples in different conditions by measuring the surface potential decay. As demonstrated in Figure 8, the surface potential decay trend is roughly the same as the filtration efficiency, indicating that the filtration efficiency decay is mainly caused by charge release. The untreated melt-blown sample has high-efficiency filtering performance, and its surface potential is −6.36 kV. The treatment temperature is the main factor affecting the decay rate of surface potential [38]. The surface potential of melt-blown sample rapidly decreased to −4.16 kV after being treated at 110 °C for 1 h. This indicates that the higher temperature could significantly accelerate the charge decay of the melt-blown material compared with the lower temperature. Furthermore, humidity also accelerates the release of charge to a certain extent [39]. The surface potential of the sample decreased to the lowest value of −1.01 kV after being treated for 24 h at 110 °C and 90% RH. These results once again show that charge escape occurs in melt-blown materials under the hot and humid environment, which results in the attenuation of filtration efficiency.

## 4. Conclusions

In summary, high temperature and high humidity treatment had no obvious effect on the fiber morphology and structure of the electret melt-blown material, so its pressure drop changed little. However, higher temperatures (90 and 110 °C) could accelerate the charge decay of the electret melt-blown material compared with the lower temperature, resulting in a significant decrease in its filtration efficiency. Besides, under the same temperature conditions, high humidity could accelerate the release of charge, so the filtration efficiency dropped more. After the sample was treated at 110 °C and 90% RH for 24 h, the filtration efficiency dropped from 95.49% to 38.16% at a flow rate of 14.16 cm s^−1^, the QF is only 0.009 Pa^−1^, and the surface potential decreased to the lowest value of −1.01 kV. Thus, during the process of production, transportation and storage of electret melt-blown nonwovens and its products, high temperature and high humidity conditions should be avoided to reduce charge decay and ensure that the final products can provide effective protection for users.

## Figures and Tables

**Figure 1 materials-13-04774-f001:**
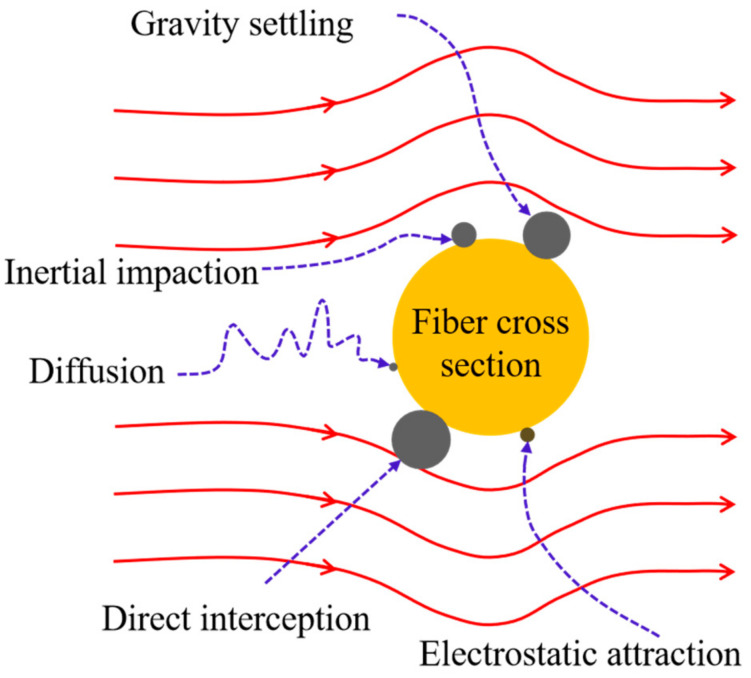
Single fiber filtration mechanism.

**Figure 2 materials-13-04774-f002:**
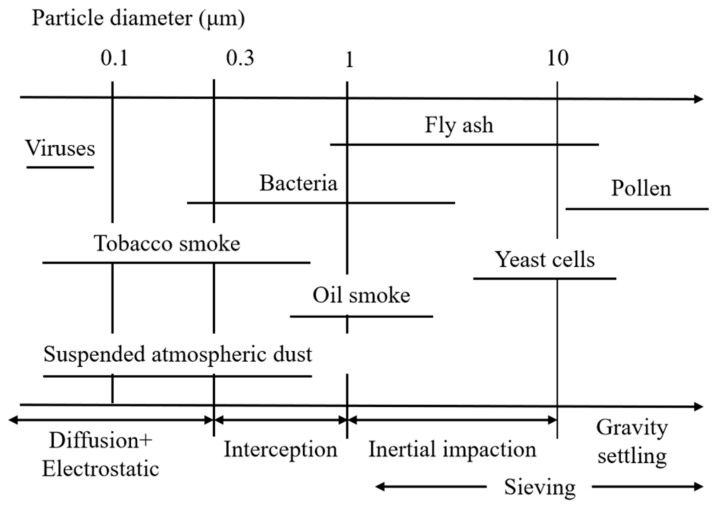
The particle size of common air pollutants corresponding to different filtration mechanisms.

**Figure 3 materials-13-04774-f003:**
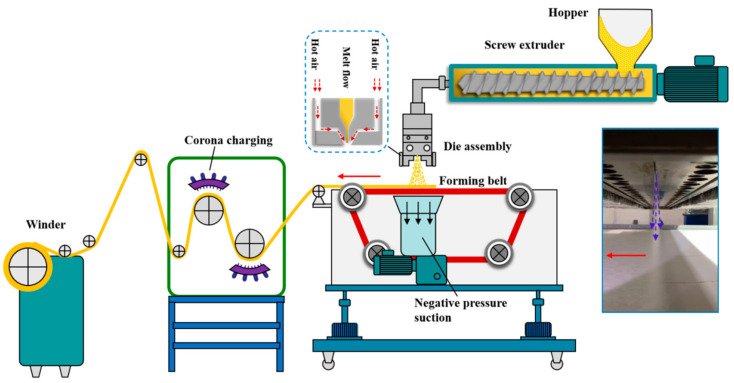
Schematic diagram of the polypropylene (PP) melt-blown nonwovens process.

**Figure 4 materials-13-04774-f004:**
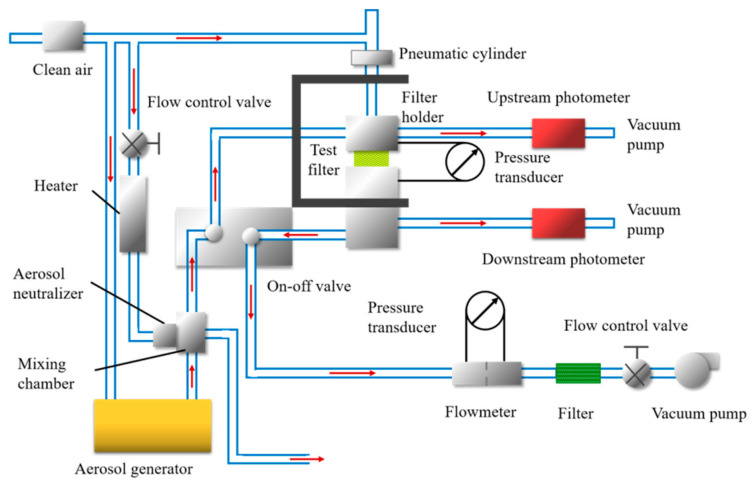
Schematic diagram of the experimental device used to test the filtration performance.

**Figure 5 materials-13-04774-f005:**
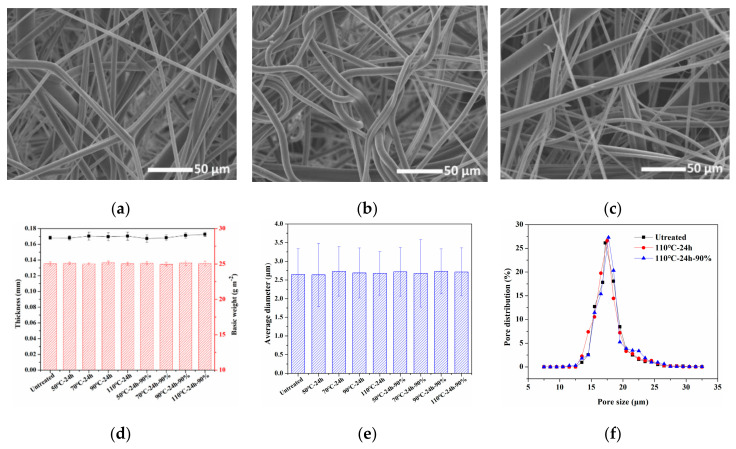
Morphology, structure, fiber diameter, and pore size of melt-blown nonwovens. SEM images of melt-blown nonwovens after various heat treatment conditions: (**a**) untreated sample, (**b**) 110 °C-24 h, and (**c**) 110 °C-24 h-90%. (**d**) Thickness and basic weight, (**e**) fiber average diameter, and (**f**) pore size distribution curves.

**Figure 6 materials-13-04774-f006:**
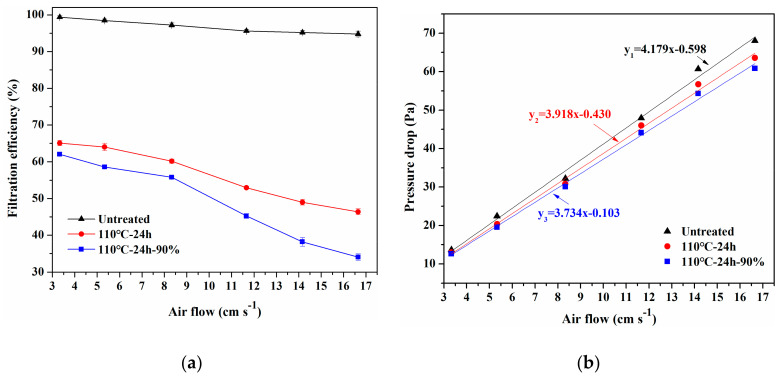
Filtration performance of melt-blown nonwovens after different heat treatment: (**a**) Filtration Efficiency (FE) and (**b**) pressure drop versus various airflow rates.

**Figure 7 materials-13-04774-f007:**
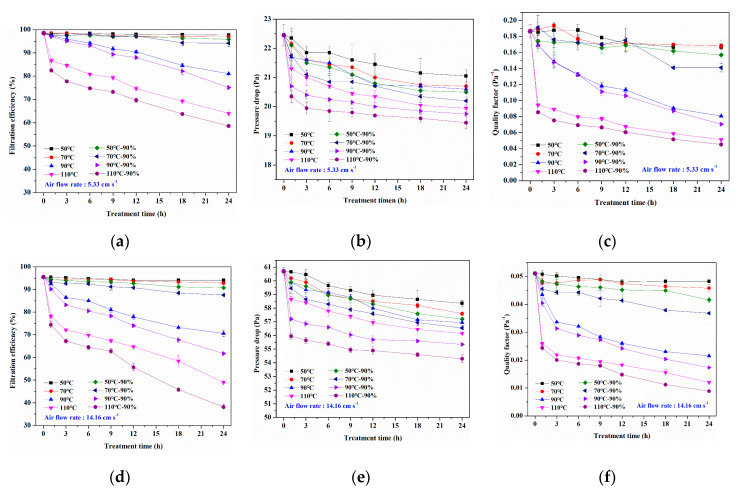
(**a**) Filtration efficiency decay, (**b**) pressure drop, and (**c**) quality factor of samples were tested at an air flow rate of 5.33 cm s^−1^. (**d**) Filtration efficiency decay, (**e**) pressure drop, and (**f**) quality factor value of samples were tested under an air flow rate of 14.16 cm s^−1^.

**Figure 8 materials-13-04774-f008:**
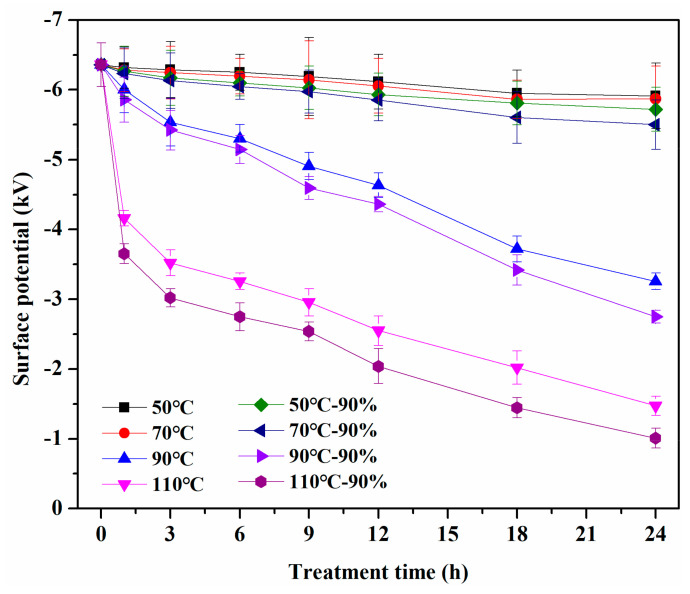
Surface potential decay of melt-blown samples treated by different conditions.

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
