# Peer review of "The Effect of Temperature and Humidity on the Filtration Performance of Electret Melt-Blown Nonwovens"

_materials, 2020, doi:10.3390/ma13214774_

Round 1
Reviewer 1 Report
Figure 4, page 5:
- it is not visible where the pneumatic cylinder is located. What is the function of a pneumatic cylinder?
- the Upstream photometer shown in the figure is not explained in the text.
- explain the upstream photometer function in the text. Explain how to measure electrostatic potential characterized by a non-contacting voltmeter.
Author Response
Dear Editors and Reviewers:
Thanks for reviewing our manuscript entitled “Effect of Temperature and Humidity on the Filtration Performance of Electret Melt-blown Nonwovens” (ID: 973722) and giving such positive view. We also thank the reviewers for all valuable comments and we believe they are helpful in improving the quality of our manuscript. We have gone over the comments, fully addressed the concerns of the reviewers and made a point-by-point response to the comments. Please see the attachement.

Reviewer 2 Report
This is a rightly written paper. It is interesting for readers although it does not bring any scientific breakthrough. I appreciate thoroughly described theory, experiments, device and the scheme of experiments. The paper is significnt for utilization of electret filters.
In the line 26, I would prefer to express the ar velocity in M/s instead of L/min.
In the Chapter 3.3. you use the terms "Filtration Performance" and "filtration efficiency". These terms seem to be used for the same phenomenon which may be mesleading for the reader. I literature, the term FE is more common.
Author Response
Response to Reviewer 2 Comments
Point 1: In the line 26, I would prefer to express the air velocity in M/s instead of L/min.
Response 1: As suggested by the reviewer, the air flow rate of 32 and 85 L/min have been converted into 5.33 and 14.16 cm/s in the revised manuscript.
Point 2: In the Chapter 3.3. you use the terms "Filtration Performance" and "filtration efficiency". These terms seem to be used for the same phenomenon which may be mesleading for the reader. I literature, the term FE is more common.
Response 2: Generally, "filtration efficiency" and "pressure drop" are two values which can be used to evaluated the overall performance of filters in air filtration. In this case, we expect "filtration performance" can be used to describe both of "filtration efficiency" and "pressure drop" of the filters against aerosols.